# Epidemiological Characteristics of Hospitalized Patients with Moderate versus Severe COVID-19 Infection: A Retrospective Cohort Single Centre Study

**DOI:** 10.3390/diseases10010001

**Published:** 2021-12-23

**Authors:** Faryal Khamis, Salah Al Awaidy, Muna Al Shaaibi, Mubarak Al Shukeili, Shabnam Chhetri, Afra Al Balushi, Sumaiya Al Sulaimi, Amal Al Balushi, Ronald Wesonga

**Affiliations:** 1Adult Infectious Diseases, Department of Medicine, Royal Hospital, Muscat PC 111, Oman; khami001@gmail.com (F.K.); Drshabnamchhetri24@gmail.com (S.C.); 2Office of Health Affairs, Ministry of Health, Muscat PC 100, Oman; 3Department of Statistics, College of Science, Sultan Qaboos University, Muscat PC 123, Oman; m.shaaibi@gmail.com (M.A.S.); s29026@student.squ.edu.om (M.A.S.); wesonga@squ.edu.om (R.W.); 4Acute Medicine, Department of Medicine, Royal Hospital, Muscat PC 111, Oman; afraa.albalushi@gmail.com (A.A.B.); samasoomi@hotmail.com (S.A.S.); ammooolah@hotmail.com (A.A.B.)

**Keywords:** COVID-19, SARS-CoV-2, demographic factors, epidemiological factors, Oman

## Abstract

COVID-19 has a devastating impact worldwide. Recognizing factors that cause its progression is important for the utilization of appropriate resources and improving clinical outcomes. In this study, we aimed to identify the epidemiological and clinical characteristics of patients who were hospitalized with moderate versus severe COVID-19 illness. A single-center, retrospective cohort study was conducted between 3 March and 9 September 2020. Following the CDC guidelines, a two-category variable for COVID-19 severity (moderate versus severe) based on length of stay, need for intensive care or mechanical ventilation and mortality was developed. Data including demographic, clinical characteristics, laboratory parameters, therapeutic interventions and clinical outcomes were assessed using descriptive and inferential analysis. A total of 1002 patients were included, the majority were male (*n* = 646, 64.5%), Omani citizen (*n* = 770, 76.8%) and with an average age of 54.2 years. At the bivariate level, patients classified as severe were older (Mean = 55.2, SD = 16) than the moderate patients (Mean = 51.5, SD = 15.8). Diabetes mellitus was the only significant comorbidity potential factor that was more prevalent in severe patients than moderate (*n* = 321, 46.6%; versus *n* = 178, 42.4%; *p* < 0.001). Under the laboratory factors; total white cell count (WBC), C-reactive protein (CRP), Lactate dehydrogenase (LDH), D-dimer and corrected calcium were significant. All selected clinical characteristics and therapeutics were significant. At the multivariate level, under demographic factors, only nationality was significant and no significant comorbidity was identified. Three clinical factors were identified, including; sepsis, Acute respiratory disease syndrome (ARDS) and requirement of non-invasive ventilation (NIV). CRP and steroids were also identified under laboratory and therapeutic factors, respectively. Overall, our study identified only five factors from a total of eighteen proposed due to their significant values (*p* < 0.05) from the bivariate analysis. There are noticeable differences in levels of COVID-19 severity among nationalities. All the selected clinical and therapeutic factors were significant, implying that they should be a key priority when assessing severity in hospitalized COVID-19 patients. An elevated level of CRP may be a valuable early marker in predicting the progression in non-severe patients with COVID-19. Early recognition and intervention of these factors could ease the management of hospitalized COVID-19 patients and reduce case fatalities as well medical expenditure.

## 1. Introduction

The 2019 Coronavirus Disease (COVID-19) pandemic of a novel coronavirus, severe respiratory distress syndrome (SARS-CoV-2), whose emergence in December 2019 from a source yet to be identified, has had a rapid expansion globally with disastrous effects. Over the past 22 weeks, the pandemic continues to progress causing more than 200 million infections and 4 million deaths globally [1]. In Oman, the first case of COVID-19 was diagnosed on 24 February 2020 [2,3].

The COVID-19 pandemic has created an extraordinary challenge to humanity worldwide due to many unknowns in understanding the transmission, genetic predisposition, disease severity risk factors and lack of efficient treatments [4]. Moreover, the disease has a wide spectrum of clinical manifestations ranging from mild to severe. An emerging public health priority, particularly under the circumstances of a novel pathogen, is to be able to predict the disease severity and risk factors associated with progression among patients. This would not only save the lives of individuals by using early and effective management strategies, but notably preserves the health systems resilience in situations where medical resources are inadequate or lacking [5].

Several factors that are associated with severe COVID-19 have been used to develop prediction models for estimating prognosis in COVID-19 patients. Comorbidities, including chronic respiratory disease, cardiovascular disease, diabetes mellitus, hypertension and immune parameters such as elevated leukocyte and neutrophil levels and reduced lymphocyte levels were observed more frequently among patients with severe COVID-19 than with the mild ones [6,7,8].

Some of these factors have a stronger disease severity association than others. Therefore, identifying risk factors in different clinical settings and populations is essential. Furthermore, knowing the severity of an illness may affect the distribution and allocation of medical staff, equipment, financial support and vaccine prioritization as the pandemic continues.

The main aim of this study was to identify the epidemiological and clinical characteristics of patients who are hospitalized with COVID-19 so as to determine the predictors of severity.

## 2. Methods

### 2.1. Study Setting

Oman is one of the 22 countries in the Eastern Mediterranean Region (EMR) of the World Health Organization (WHO) with a population of nearly 5 million, out of whom approximately 43% are foreign-born individuals (non-citizen) [9]. The Royal Hospital (RH) is the largest tertiary health care facility with nearly 1000 beds and advanced critical care units.

### 2.2. Study Design and Population

We conducted a single-center, retrospective cohort study at the RH. The hospital has been designated as a major hospital for admissions of patients with a clinical spectrum of moderate, severe and critical COVID-19 as follows:

Moderate Illness: Individuals who show evidence of lower respiratory disease during clinical assessment or imaging and who have an oxygen saturation (SpO_2_) ≥ 94% on room air at sea level.

Severe Illness: Individuals who have SpO_2_ < 94% on room air at sea level, a ratio of arterial partial pressure of oxygen to fraction of inspired oxygen (PaO_2_/FiO_2_) < 300 mm Hg, a respiratory rate > 30 breaths/min or lung infiltrates > 50%.

Critical Illness: Individuals who have respiratory failure, septic shock and/or multiple organ dysfunction.

Data for 1002 consecutive patients admitted with COVID-19 were collected from 3 March to 9 September 2020. The inclusion criteria were as follows: (1) Patients with laboratory-confirmed COVID-19 by viral nucleic acid detection using RT-PCR with samples from nasopharynx swabs; (2) Patients with a clinical spectrum of moderate, severe and critical COVID-19 disease based on U.S. Department of Health & Human Services, National Institutes of Health (NIH) criteria (www.covid19treatmentguidelines.nih.gov, accessed on 15 November 2021) who required hospitalization; (3) Patients who underwent complete laboratory tests on admission (routine blood tests, biochemistry analysis, liver functions test, CRP, LDH) and clinical recording at admission; and (4) Patients who underwent chest imaging on admission. We excluded the pediatric age group. All COVID-19 cases were categorized into two main groups: moderate and severe cases. Severe cases met at least one of the following four criteria based on CDC guidelines [10]: (1) whether the patient was admitted to ICU; (2) required mechanical ventilation; (3) hospitalized for 9 or more days; and (4) mortality outcome.

### 2.3. Data Collection and Analysis

The data collected included demographic (gender, age and nationality), clinical presentation on admission, laboratory parameters, therapeutic and patients’ outcomes (hospitalization, mortality and recovery).

### 2.4. Stastical Methods

The data were analyzed using descriptive and inferential statistics by generating frequency tables and unadjusted odds ratios at the bivariate level, while adjusted odds ratios were used at the multivariate level to identify the factors associated with COVID-19 severity. Associations between categorical variables were assessed either by using the Chi-square test whereas continuous variables were analyzed by the Student’s *t*-test and a priori two-tailed level of significance was set at 0.05. A logistic regression model was developed to determine factors associated with moderate versus severe COVID-19 illness. Only those factors with a *p*-value less than 0.05 at the bivariate level of analysis were selected for inclusion in the logistic regression analysis. We fitted a binary logistic regression model to determine the variables that are significantly associated with moderate and severe COVID-19 for hospitalized patients. The factors evaluated were categorized under demographic, comorbidities, clinical, laboratory and therapeutic interventions. The rationale for selecting binary logistic regression was its ability to identify and test the hypothesis that there is no relationship between the potential predictors and COVID-19 severity levels. Statistical analyses were conducted using R statistical packages.

### 2.5. Ethical Approval

Ethical approval for this study was obtained from the Royal Hospital Ethical and Research Committee (SR #26/2020). This study does not contravene the internal institutional review board and follows the Declaration of Helsinki.

## 3. Results

### 3.1. Description of the Characteristics of Hospitalized COVID-19 Patients (N = 1002)

In all, 1002 patients were included in the study. Among those, 420 (41.9%) were moderate and 582 (58.1%) were severe. The majority were male (*n* = 646, 64.5%), Omani citizen (*n* = 770, 76.8%) and had an average age of 54.2 (SD = 16.09) years (Table 1). Of the total cases, 51.6% (*n* = 517) were admitted in the ICU, 41.3% (*n* = 414) required mechanical ventilation and 25.7% (*n* = 257) died (Figure 1 and Table 1). The median hospital stay was 8 days (4–15) for the moderate group and 13 days (9–21) for those in severe group.

Their clinical characteristics showed that the majority suffered from diabetes (*n* = 503, 50.2%) and hypertension (*n* = 514, 51.3%). On admission, shortness of breath (*n* = 836, 83.4%) and bilateral pulmonary infiltrations (*n* = 847, 84.5%), were the two most common clinical and radiological manifestations. As for therapeutic interventions, plasmapheresis was performed in six patients (26.2%), while most of the patients (*n* = 649, 64.8%) were treated with steroids.

### 3.2. Relationship of the Potential Factors for the COVID-19 Patients with Disease Severity (N = 1002)

Table 2 shows analysis by disease severity levels, that is, moderate and severe COVID-19. The distribution of demographic factors, clinical characteristics, laboratory findings and therapeutic interventions was tested for the two levels of severity (moderate, severe) in COVID-19 patients. All demographic, clinical and therapeutic factors were significantly associated with severity. Under the comorbidities, only diabetes mellitus was a significant risk factor for severity, while high WBC, CRP, LDH, D-dimer and corrected calcium were laboratory parameters associated with severe disease.

Furthermore, patients classified as severe were significantly older (Mean = 56.0, SD = 16.0) than those who were moderate (Mean = 51.5, SD = 15.8). Our bivariate analysis identified the following risk factors to be significantly associated with severity: gender (Male: *n* = 396 (68.2%), (Female: *n* = 185 (31.8%)), nationality (Other: *n* = 417 (71.8%), Omani citizen, *n* = 164 (28.2%)), diabetes mellitus (DM) (*n* = 321, 55.2%), Shortness of breath (SOB) (*n* = 544, 93.8%), Bilateral lung consolidation on X-ray (*n* = 550, 94.5%), sepsis (*n* = 159, 27.3%), Acute respiratory distress syndrome (ARDS) (*n* = 266, 45.7%) and Non-invasive ventilation requirement (NIV) (*n* = 266, 45.7%). Similar results were observed among the therapeutic interventions, whereby the use of plasmapheresis (*n* = 60, 10.3%), convalescent plasma (*n* = 361, 43.0%, *p* < 0.05), tocilizumab (*n* = 316, 37.7%, *p* < 0.05) and steroids (*n* = 614, 73.2%, *p* < 0.05) were significantly higher among the patients classified with severe COVID-19.

### 3.3. Identifying Potential Risk Factors for Severity among Hospitalized COVID-19 Patients

Table 3 summarizes the unadjusted and adjusted odds ratios from the logistic model used to identify potential risk factors for COVID-19 severity among hospitalized patients. The unadjusted analysis corresponds with the results from our earlier parameters in Table 2. One common characteristic is that for all the demographic, clinical characteristics, laboratory and therapeutic factors, the odds ratios (ORs) are greater than one, associated with severe COVID-19.

At the adjusted level, the single significant demographic factor was nationality (OR = 0.43, 95% CI: 0.22–0.83, *p* = 0.012). Results show that being Omani reduced the odds of acquiring severe COVID-19 by 43% compared to non-Omani hospitalized patients.

None of the comorbidities were significant, however, among the clinical factors, sepsis (OR = 3.76, 95% CI: 1.35–11.20, *p* = 0.013), ARDS (OR = 27.05, 95% CI: 11.03–82.00, *p* < 0.001) and NIV (OR = 7.09, 95% CI: 2.80–20.77, *p* < 0.001) were significantly associated with severe COVID-19. Among the significant laboratory parameters only CRP (mg/L) (OR = 1.00, 95% CI: 1.00–1.01, *p* = 0.045) was identified as being significant. The total average WBC on admission for patients with COVID-19 classified as severe (Mean = 9.1, SD = 5.9) was higher than for those classified as moderate (Mean = 6.2, SD = 3.6) but this was not a significant factor. Of the four treatment factors considered in the current study, only requiring steroids (OR = 3.28, 95% CI: 1.81–6.07, *p* < 0.001) was significantly associated with severe COVID-19 among the hospitalized patients.

## 4. Discussion

The number of patients with COVID-19 continues to grow globally. About 80% of COVID-19 patients tend to have mild illness, of whom 10–20% will progress to moderate and severe disease requiring hospitalization and ICU care [11,12,13,14]. Identifying factors associated with moderate versus severe illness is key in providing the most effective management strategy for hospitalized patients and prioritizing limited resources. Several existing studies have identified risk factors mostly of mortality in patients with COVID-19 [15,16,17,18,19,20,21,22], however, little is known on factors aggravating COVID-19 symptoms. To our knowledge, this study is one of the few retrospective cohort studies comparing demographic, clinical characteristics, laboratory parameters and therapeutic interventions in hospitalized COVID-19 patients with moderate versus severe disease. The stratification of COVID-19 severity was based on CDC guidelines that included: need for ICU admission, need for MV, length of stay (LOS) for 9 or more days and mortality in an attempt to identify factors leading to progression of COVID-19 from one clinical stage into another.

Among 1002 admitted patients, we found an association between several risk factors and disease severity that include: age, nationality, male gender, presence of DM, shortness of breath (SOB) on presentation, bilateral chest consolidations, ARDS, sepsis, high WBC, high CRP, high LDH, high D-dimer, low corrected calcium and need for NIV. On adjusted level, non-Omani nationality, ARDS, sepsis, high CRP and need for NIV and steroids use were associated with severe disease. For example, the odds of receiving steroids were 3.28 (1.81–6.07, *p* < 0.001) times more likely in patients with severe than moderate COVID-19 among the hospitalized patients at the Royal Hospital.

Gender differences have been shown to be a risk factor for mortality in COVID-19 [23,24]. Similarly in our study, males were found to be at a higher risk for developing severe disease. On the contrary, few other studies [25,26] have suggested that the effect of gender and ethnicity on progression of COVID-19 is not significant. The observed gender differences for COVID-19 mortality are likely attributed to a combination of factors such as hormonal differences, immune response and surge of inflammatory markers. Additionally, differences in attitudes and behaviors also impact infection and outcomes [26].

In this study, being non-national was associated with severe disease. It could be related to the ethnicity and race [24,25,26] but also might be related to the access to care and availability of health services. In matter of fact, all Omanis are provided with health care services free of charge at all governmental institutions, while non-Omanis seek paid health services in the private sector. The delay in accessing healthcare services may explain the detrimental impact of COVID-19 on this group.

Several studies reported age to be the most important predictor of severity and death in patients with COVID-19 [15,17]. Similarly, we observed a tendency for older patients to progress from moderate to severe-stage disease. The age-dependent functional defects in immunologic cells lead to impaired suppression of viral replication [27,28,29]. Furthermore, the prevalence rate of comorbidities among older people tends to be high [30]. However, in our multivariate analysis and in concurrence with some other reports, age was not a predictive factor for disease severity. This inconsistency may be explained by the relatively young cohort assessed in our study. Additionally, we studied risk factors for the COVID-19 hospitalized patients classified as moderate versus severe and not only mortality.

In recent clinical reports on COVID-19 derived from symptomatic hospitalized patients, older age and underlying comorbidities such as DM, hypertension, cardiovascular disease and liver disease have been identified as risk factors for severe disease [2,31,32]. Similarly, in our cohort, DM was the comorbidity most associated with severity. The high risk of progression to severe COVID-19 in patients with DM is likely as hyperglycemia potentially triggers immune impairment affecting neutrophil function, antioxidant system function, humoral immunity [33,34,35,36] and increases the susceptibility to nosocomial infections [35]. Therefore, intensive treatment for DM should be considered when managing patients with COVID-19. In some studies [18,32,33] an increase in the severity of COVID-19 was reported among patients with hypertension, chronic respiratory conditions, hyperlipidemia and chronic liver disease, however, these factors were not statistically significant in our setting, although a higher trend was observed among patients with severe COVID-19. A meta-analysis by Yadav et al. showed that COVID-19 patients with liver disease had an increased risk of progression by 53.5% and mortality by 23.5% [37]. The angiotensin II-converting enzyme (ACE2) receptor, which has an essential role in binding and replication of the SARS-CoV-2 virus, is expressed by cholangiocytes suggesting that the binding of the virus to the biliary epithelium causes biliary dysfunction [38,39].

As other studies have shown [40,41,42], SOB and the finding of bilateral consolidations on chest radiography upon initial presentation are indicative of substantial effects of the virus, signifying the likelihood of progression from moderate to severe conditions in patients with these abnormal findings. However, early in the progression of this disease, radiographic findings might be normal in 15% of individuals undergoing CT imaging and among 40% of individuals undergoing chest radiography [43].

In this study, the clinical parameter to assess sepsis on initial presentation was the qSOFA [44]. Viral sepsis has been suggested as a term to describe the multisystem dysregulations and clinical findings in severe and critically ill COVID-19 patients where there is an interplay between hyperinflammation and altered metabolism. An integrated host-dependent dysregulation of inflammatory cytokines, neutrophil activation chemokines, glycolysis, mitochondrial metabolism, amino acid metabolism, polyamine synthesis and lipid metabolism typical of sepsis processes are seen more often in severe COVID-19 disease and are indicative of progression and worse outcome [45].

Likewise, ARDS is a predictor for severe and even fatal COVID-19. ARDS develops in 42% of patients with COVID-19 pneumonia, and 61%–81% of patients requiring ICU care [46]. COVID-19 ARDS follows a predictable time course over days, with a median time to intubation of 8.5 days after symptom onset [47]. It is therefore essential to monitor respiratory rate and SpO_2_ to identify patients with ARDS early in order to prevent COVID-19 disease progression. The mortality in COVID-19 ARDS ranges between 26% and 61.5% if the patient requires admission into ICU, and in patients on MV, the mortality ranged between 65.7% and 94% [46].

COVID-19 ARDS is characterized by the typical ARDS pathological changes of diffuse alveolar damage in the lung that can progress to lung fibrosis due to the gradual replacement of cellular components by scar tissues. In fatal cases there is diffuse microvascular thrombosis, suggesting a thrombotic microangiopathy and disseminated intravascular coagulation [48]. Thus, early monitoring of changes in coagulation cascade including D-dimer is important.

In this study, there were significant differences in the initial laboratory parameters between the two groups in total WBC, CRP, LDH, D-dimer and corrected calcium. Higher levels of these laboratory parameters were found in the severe group compared with the moderate group. The severity of COVID-19 infection may stimulate neutrophils to generate an immune response to the virus that gets dysregulated and causes a profound “cytokine storm” [49]. In several other reports, low lymphocyte count and high inflammatory markers levels were also associated with increased severity [50]. Further studies are required to determine if these laboratory markers can be used as a proxy for disease severity.

Interestingly, hypocalcemia has been reported by several researchers as being predictive of poor clinical outcome in hospitalized critically ill patients [51,52]. Furthermore, calcium has been found to be pivotal in the viral fusion of enveloped viruses such as SARS-CoV, MERS-CoV and Ebola viruses, enhancing their replication [53,54]. Likewise, a high incidence of hypocalcemia has been observed in COVID-19 patients and was suggested to be a reliable predictor for the need for hospitalization and disease progression. The detrimental effect of hypocalcemia in COVID-19 patients might be due to its negative impact on cardiac function that can be fatal if severe or acute [55], and its roles in coagulation and inflammation cascades. Patients with COVID-19 tend to have high levels of unbound fatty acids and unsaturated fatty acids with a proinflammatory role that bind to calcium causing significant acute hypocalcemia and promoting cytokine storm, multisystem organ failure and acute lung injury [56].

In this cohort, the odds ratio of high CRP was more than one for the patients classified as severe COVID-19 disease rather than moderate. An elevated level of CRP may be an early marker to predict disease progression in patients with moderate COVID-19, and might help clinicians in identifying patients at risk for progression. The high levels of CRP could be due to the overproduction of inflammatory cytokines and tissue destruction in severe COVID-19 [57].

Furthermore, in our adjusted analysis, among patients requiring NIV, the odds of disease severity were 7.09 (2.80–20.77, *p* < 0.001) times higher in severe than moderate COVID-19 hospitalized patients. In their sub-analysis of a prospective multinational registry of critically ill COVID-19 patients, Wendel Garcia and colleagues [58] compared NIV to the other respiratory support strategies such as standard oxygen therapy (SOT), high flow nasal cannula (HFNC) and invasive mechanical ventilation (IMV). The use of NIV was associated with a higher overall ICU mortality (SOT: 18%, HFNC: 20%, NIV: 37%, IMV: 25%, *p* = 0.016). The median duration of the in-hospital stay until intubation and the ICU length of stay were longer in the NIV group than in the other three groups. These findings have been reported in patients with ARDS, supporting that NIV should be avoided due to the increased risk of ICU death [59]. In ARDS of viral etiology, the use of NIV is associated with high failure rates up to 85% [60].

Based on our “National Protocol for Management of Hospitalized Patients”, therapies such as convalescent plasma (CP), tocilizumab and steroids were used early on in moderate and severe COVID-19 patients to prevent further progression. In this cohort, our results at the bivariate level show that all the proposed therapies had significant association with severity and had an impact on disease progression.

In our study, 35% of patients in the moderate group received steroids versus 73% in the severe group. The multivariant analysis indicated that steroids were 3.28 (95% CI: 1.81–6.07, *p* < 0.05) times more likely to be used as an intervention in severe than moderate COVID-19 hospitalized patients to prevent further progression. Both observational studies and RCTs confirmed a beneficial effect of using low dose corticosteroids in severe COVID-19 pneumonia on short-term mortality, disease progression, need for mechanical ventilation and LOS [61,62,63,64,65,66,67,68,69,70].

The use of CP collected from infected individuals and passively transferring antibodies to others in order to protect or treat has been practiced for more than 100 years. Prior research on MERS and SARS coronavirus outbreaks suggested that CP is safe and may confer clinical benefits, including faster viral clearance, particularly when administered early in the disease course [71,72,73]. In some recent studies, early use of CP has been shown to decrease progression of COVID-19 illness [72,73,74,75,76,77], however, the reports on the benefit of CP in COVID-19 have been inconsistent [78,79,80,81,82,83,84,85,86]. In this study, 19.6% of patients with moderate disease received CP early into the illness. On adjusted analysis, the use of CP was not found to be significantly associated with disease severity.

Similarly, among this cohort, 37.7% of patients received tocilizumab in the severe group versus only 5.5% in the moderate group. Tocilizumab was given per our protocol for Management of COVID-19 in patients with evidence of cytokine storm and at risk of increasing ventilatory requirements [2]. Early tocilizumab use with signs of increasing oxygen requirement was found to be beneficial in preventing disease progression. In EMPACTA trial, nonventilated hospitalized patients requiring oxygen who received tocilizumab in the first 2 days of admission had a lower risk of progression to MV or death by day 28 compared with those not treated with tocilizumab (12% vs. 19.3% respectively) [86]. Furthermore, in Recovery trial, among those not receiving IMV at baseline, patients who received tocilizumab were less likely to reach the composite endpoint of IMV or death (35% vs. 40%) [87].

## 5. Strengths and Limitations

One of the important strengths of this study is the enrollment of moderate and severe COVID-19 cases confirmed by RT-PCR who were admitted to a single center, undergoing the same investigations and treatment protocols. This allowed accurate determination of the proportions of moderate cases and severe cases. Moreover, the relatively large sample size of patients and the wide spectrum of clinical disease identifies risk factors for severity with great precision.

There were few limitations in this study. First, the retrospective nature of the study and missing data precluded detailed analyses of other significant factors associated with disease progression. Second, some important risk factors reported in previous research such as comorbidities and some signs and symptoms showed no significant difference between the two groups in multivariate analysis. A larger sample is needed to show the differences, however, since we included as many COVID-19 cases as possible in our hospital, we believe the study population is representative of cases diagnosed and treated. Thirdly, the changes in patient characteristics from admission to outcome were not addressed as clinical assessments and laboratory tests were performed at admission but not on discharge. Information on temporal changes could further ascertain risk factors for disease progression. Finally, data in our study were from March to September 2020 and the epidemiological and clinical characteristics might be different now with the mutation of the virus and emergence of the delta variant in the country thus, generalizations to other countries should be made with caution.

## 6. Conclusions

We evaluated the risk factors for developing moderate versus severe COVID-19 illness among hospitalized patients. Factors including age, gender, nationality, DM, SOB and/or bilateral lung consolidation on admission, ARDS and sepsis are associated with severe COVID-19 among hospitalized patients. Moreover, high WBC, CRP, LDH, D-dimer, low corrected calcium and need for NIV are parameters significantly associated with severe COVID-19. COVID-19 patients with elevated levels of CRP need close monitoring and early treatment even in the absence of severe disease as risk of progression is significant. Early recognition of severity factors could not only ease the management of hospitalized COVID-19 patients but may also reduce case fatalities and medical expenditure. Further large-scale multicenter studies are warranted.

## Figures and Tables

**Figure 1 diseases-10-00001-f001:**
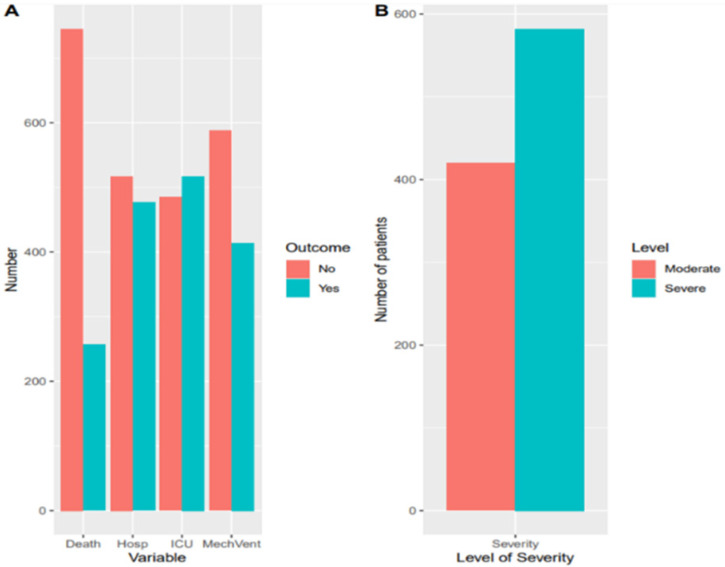
(**A**) COVID-19 outcomes. (**B**) Severity factors for moderate versus severe COVID-19 patients.

**Table 1 diseases-10-00001-t001:** Descriptive characteristics of the COVID-19 hospitalized patients (*N* = 1002).

Factors	Measurement	*N*	Percent (%)
**Severity Outcome**			
Died	Yes	257	25.7
Hospitalized at least 9 days	Yes	477	47.6
Admitted to ICU	Yes	517	51.6
Required mechanical ventilation	Yes	414	41.3
**Demographics**			
Age *	Mean (SD)	54.13 (16.09)	
Gender	Male	355	35.4
	Female	646	64.5
Nationality	Others	231	23.1
	Omani	770	76.8
**Comorbidities**			
Diabetes Mellitus	No	503	50.2
	Yes	499	49.8
Hypertension	No	488	48.7
	Yes	514	51.3
Dyslipidemia	No	786	78.4
	Yes	216	21.6
Respiratory disease	No	886	88.4
	Yes	116	11.6
Cardiac disease	No	793	79.1
	Yes	209	20.9
Liver disease	No	959	95.7
	Yes	43	4.3
**Clinical Factors**			
Shortness of breath	No	162	16.2
	Yes	836	83.4
Bilateral lung consolidation	No	155	15.5
	Yes	847	84.5
Sepsis	No	828	82.6
	Yes	174	17.4
Acute respiratory distress syndrome	No	577	57.6
	Yes	425	42.4
Non-invasive ventilation requirement	No	722	72.1
	Yes	280	27.9
**Laboratory Parameters**			
Total white cell count, (10^9^/L) *	Mean (SD)	7.7 (4.7)	
Hemoglobin A1c,(mmol/L) *	Mean (SD)	1.2 (1.1)	
C-reactive protein(mg/L) *	Mean (SD)	106.5 (82.1)	
Lactate dehydrogenase (mmol/L) *	Mean (SD)	470.65 (536.5)	
Ferritin, (ng/mL) *	Mean (SD)	1283.45 (2410.2)	
Alanine aminotransferase, (U/L) *	Mean (SD)	65.15 (167.1)	
D-dimer, (ng/mL) *	Mean (SD)	4.45 (13.3)	
Corrected calcium, (mmol/L) *	Mean (SD)	2.15 (0.2)	
Troponin (pg/mL) *	Mean (SD)	159.1 (938.2)	
Vitamin D (IU) *	Mean (SD)	69.4 (36.9)	
Estimated glomerular filtration rate (mL/min/1.73 m^2^)	No	689	68.8
	Yes	311	31.0
**Therapeutic Interventions**			
Plasmapheresis	No	940	93.8
	Yes	62	6.2
Convalescent plasma	No	609	60.8
	Yes	393	39.2
Tocilizumab	No	677	67.6
	Yes	325	32.4
Steroids	No	353	35.2
	Yes	649	64.8

* The numeric factor where mean and standard deviation (SD) are presented.

**Table 2 diseases-10-00001-t002:** Factors associated with severity of COVID-19 hospitalized patients (*N* = 1002).

Factor	Measurement	Severity Level		*p*-Value
		Moderate *N* = 163	Severe *N* = 839	
**Demographic Factors**				
Age	Mean (SD)	51.5 (15.8)	56.0 (16.0)	<0.001
Gender	Male	250 (59.5)	396 (68.2)	0.006
	Female	170 (40.5)	185 (31.8)	
Nationality	Omani citizen	353 (84.0)	417 (71.8)	<0.001
	Others	67 (16.0)	164 (28.2)	
**Comorbidities**				
Diabetes Mellitus	No	242 (57.6)	261 (44.8)	<0.001
	Yes	178 (42.4)	321 (55.2)	
Hypertension	No	216 (51.4)	272 (46.7)	0.161
	Yes	204 (48.6)	310 (53.3)	
Dyslipidemia	No	340 (81.0)	446 (76.6)	0.118
	Yes	80 (19.0)	136 (23.4)	
Respiratory disease	No	378 (90.0)	508 (87.3)	0.220
	Yes	42 (10.0)	74 (12.7)	
Cardiac disease	No	344 (81.9)	449 (77.1)	0.080
	Yes	76 (18.1)	133 (22.9)	
Liver disease	No	407 (96.9)	552 (94.8)	0.153
	Yes	13 (3.1)	30 (5.2)	
**Clinical Factors**				
Shortness of breath	No	126 (30.1)	36 (6.2)	<0.001
	Yes	292 (69.9)	544 (93.8)	
Bilateral lung consolidation	No	123 (29.3)	32 (5.5)	<0.001
	Yes	297 (70.7)	550 (94.5)	
Sepsis	No	405 (96.4)	423 (72.7)	<0.001
	Yes	15 (3.6)	159 (27.3)	
Acute respiratory distress syndrome	No	408 (97.1)	169 (29.0)	<0.001
	Yes	12 (2.9)	413 (71.0)	
Non-invasive ventilation requirement	No	406 (96.7)	316 (54.3)	<0.001
	Yes	14 (3.3)	266 (45.7)	
**Laboratory Parameters**				
Total white cell count, (10^9^/L)	Mean (SD)	6.2 (3.6)	9.1 (5.9)	<0.001
Hemoglobin A1C, (mmol/L)	Mean (SD)	1.2 (0.7)	1.2 (1.4)	0.207
C-reactive protein(mg/L)	Mean (SD)	81.8 (70.9)	129.1 (91.7)	<0.001
Lactate dehydrogenase, (mmol/L)	Mean (SD)	403.0 (302.5)	534.1 (463.8)	<0.002
Ferritin, (ng/mL)	Mean (SD)	1205.0(1857.3)	1391.9 (3020.1)	0.477
Alanine aminotransferase, (U/L)	Mean (SD)	60.0 (81.7)	71.7 (252.1)	0.576
D-dimer,(ng/mL)	Mean (SD)	2.5 (9.9)	6.4 (16.9)	0.013
Corrected calcium, (mmol/L)	Mean (SD)	2.1 (0.1)	2.2 (0.2)	<0.001
Troponin, (pg/mL)	Mean (SD)	64.6 (224.7)	227.9 (1532.5)	0.370
Vitamin D, (IU)	Mean (SD)	66.5 (36.3)	72.0 (37.6)	0.382
Estimated glomerular filtration rate (mL/min/1.73 m^2^)	No	287 (68.5)	402 (69.2)	0.869
	Yes	132 (31.5)	179 (30.8)	
**Therapeutic Interventions**				
Plasmapheresis	No	418 (99.5)	522 (89.7)	<0.001
	Yes	2 (0.5)	60 (10.3)	
Convalescent plasma	No	131 (80.4)	478 (57.0)	<0.001
	Yes	32 (19.6)	361 (43.0)	
Tocilizumab	No	154 (94.5)	523 (62.3)	<0.001
	Yes	9 (5.5)	316 (37.7)	
Steroids	No	128 (78.5)	225 (26.8)	<0.001
	Yes	35 (21.5)	614 (73.2)	

**Table 3 diseases-10-00001-t003:** Demographic, clinical, laboratory and therapeutic determinants of COVID-19 severity among hospitalized patients.

Factor	Measure	Severity Level		Unadjusted OR (95% CI, P)	Adjusted OR (95% CI, P)
		Moderate (*N* = 163)	Severe (*N* = 839)		
**Demographics**					
Age	Mean (SD)	51.5 (15.8)	56.0 (16.0)	1.02 (1.01–1.03, *p* < 0.001)	1.00 (0.98–1.02, *p* = 0.961)
Gender	Female	170 (47.9)	185 (52.1)	-	-
	Male	250 (38.7)	396 (61.3)	1.46 (1.12–1.89, *p* = 0.005)	1.46 (0.80–2.70, *p* = 0.225)
Nationality	Others	67 (29.0)	164 (71.0)	-	-
	Omani	353 (45.8)	417 (54.2)	0.48 (0.35–0.66, *p* < 0.001)	0.43 (0.22–0.83, *p* = 0.012)
**Comorbidities**					
Diabetes Mellitus	No	242 (57.6)	261 (44.8)	-	-
	Yes	178 (42.4)	321 (55.2)	1.67 (1.30–2.16, *p* < 0.001)	0.99 (0.55–1.78, *p* = 0.986)
**Clinical Factors**					
Shortness of breath	No	126 (30.1)	36 (6.2)	-	-
	Yes	292 (69.9)	544 (93.8)	6.52 (4.43–9.82, *p* < 0.001)	1.20 (0.47–3.33, *p* = 0.711)
Bilateral lung infiltrates	No	123 (29.3)	32 (5.5)	-	-
	Yes	297 (70.7)	550 (94.5)	7.12 (4.76–10.92, *p* < 0.001)	2.38 (0.83–7.45, *p* = 0.118)
Sepsis	No	405 (96.4)	423 (72.7)	-	-
	Yes	15 (3.6)	159 (27.3)	10.15 (6.07–18.25, *p* < 0.001)	3.76 (1.35–11.20, *p* = 0.013)
Acute respiratory distress syndrome	No	408 (97.1)	169 (29.0)	-	-
	Yes	12 (2.9)	413 (71.0)	83.09 (47.53–159.85, *p* < 0.001)	27.05 (11.03–82.00, *p* < 0.001)
Non-invasive ventilation requirement	No	406 (96.7)	316 (54.3)	-	-
	Yes	14 (3.3)	266 (45.7)	24.41 (14.50–44.57, *p* < 0.001)	7.09 (2.80–20.77, *p* < 0.001)
**Laboratory Parameters**					
Total white cell count, (10^9^/L)	Mean (SD)	6.2 (3.6)	9.1 (5.9)	1.14 (1.11–1.18, *p* < 0.001)	1.02 (0.97–1.08, *p* = 0.355)
C-reactive protein (mg/L)	Mean (SD)	81.8 (70.9)	129.1 (91.7)	1.01 (1.00–1.01, *p* < 0.001)	1.00 (1.00–1.01, *p* = 0.045)
Lactate dehydrogenase, (mmol/L)	Mean (SD)	403.0 (302.5)	534.1 (463.8)	1.00 (1.00–1.00, *p* < 0.001)	1.00 (1.00–1.00, *p* = 0.162)
D-dimer, (ng/mL)	Mean (SD)	2.5 (9.9)	6.4 (16.9)	1.04 (1.02–1.06, *p* = 0.001)	1.00 (0.98–1.03, *p* = 0.781)
Corrected calcium, (mmol/L)	Mean (SD)	2.1 (0.1)	2.2 (0.2)	1.06 (0.55–2.05, *p* = 0.859)	2.05 (0.45–9.58, *p* = 0.352)
**Therapeutic Interventions**					
Plasmapheresis	No	418 (99.5)	522 (89.7)	-	-
	Yes	2 (0.5)	60 (10.3)	24.02 (7.45–147.04, *p* < 0.001)	2.56 (0.55–19.47, *p* = 0.283)
Convalescent plasma	No	131 (80.4)	478 (57.0)	-	-
	Yes	32 (19.6)	361 (43.0)	5.28 (3.94–7.13, *p* < 0.001)	1.18 (0.63–2.17, *p* = 0.608)
Tocilizumab	No	154 (94.5)	523 (62.3)	-	-
	Yes	9 (5.5)	316 (37.7)	6.83 (4.90–9.68, *p* < 0.001)	1.15 (0.59–2.23, *p* = 0.680)
Steroids	No	128 (78.5)	225 (26.8)	-	-
	Yes	35 (21.5)	614 (73.2)	6.28 (4.73–8.38, *p* < 0.001)	3.28 (1.81–6.07, *p* < 0.001)

## Data Availability

The data will be shared upon reasonable request.

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
