# Peer review of "Epidemiological Characteristics of Hospitalized Patients with Moderate versus Severe COVID-19 Infection: A Retrospective Cohort Single Centre Study"

_diseases, 2021, doi:10.3390/diseases10010001_

Round 1
Reviewer 1 Report
The Ms of Faryal Khamis and coworkers aimed to identify the epidemiological and clinical characteristics of patients who were hospitalized with moderate versus severe COVID-19 illness. They conclude various parameters (i.e. higher WCB, low corrected calcium, required NIV and steroids) are significantly associated with increased odds of developing severe COVID.19 among hospitalized patients.
The MS required moderate English changes.
The MS could be accepted in the present form.
Author Response
The responses to reviewer #1 are attached

Reviewer 2 Report
The authors collected data from a retrospective study (n=1002) at RH in Oman, and examined the factors associated with severity of covid-19. The study has a strength in expanding the global research efforts to Oman people. However, it is cautious that the authors tend to interpret the result as the progression to severe covid illness, despite that the outcome variable (and inclusion criteria) was not defined as the progression. In addition, the multiple logistic regression poses concerns in identifying risk factors for increased odds of severe covid, as some relationships such as between therapeutic interventions or clinical factors with the severity might have been driven by the outcome-dependent interventions (e.g., admitted at ICU may promote steroid use).
- Can you justify including ‘hospitalization for 9 or more days’ as one of criteria for the outcome variable (covid 19 severity) in lines 99-101? What’s the range and median of hospital stay in overall, and stratified by death? How many patients meeting each criteria for the outcome variable?
- Add units for variables in Tables - units for HbA1c (%? mmol/L?).
- In Analytical statistics subsection, add a description on which variables were selected to be included in a logistic regression.
- Is the data on clinical factors (shortness of breath, etc) measured when admitted in the hospital or any occurrence during the hospital stay?
- If it is possible to re-configure the data to run a prospective association analysis (developing severe covid from mild), the manuscript will be improved a lot.
Author Response
The responses to reviewer #2 are enclosed

Reviewer 3 Report
The work is well written, in line with recent literature (present in the bibliography).
In my opinion, the following changes are necessary:
Include the units of measurement in the various Tables when required;
do you have mortality data available?
Considering the WBC, is it possible that this data is affected by a larger use of steroids?
Can we consider NIV "a significant predictor for severe COVID-19" considering that the use of NIV is a direct consequence of respiratory failure rather than a predictor?
Author Response
The responses for reviewer #3 are enclosed

Round 2
Reviewer 2 Report
comments were well addressed
Reviewer 3 Report
Authors well answer to my questions, resolving my doubt.